# Transcranial Direct-Current Stimulation Improves Verbal Fluency in Children with Attention Deficit Hyperactivity Disorder (ADHD)

**DOI:** 10.3390/brainsci13091257

**Published:** 2023-08-29

**Authors:** Vahid Nejati, Reza Estaji, Zahra Helisaz

**Affiliations:** Department of Psychology, Shahid Beheshti University, Tehran P.O. Box 1983969411, Iran

**Keywords:** attention deficit hyperactivity disorder (ADHD), dorsolateral prefrontal cortex (dlPFC), inferior frontal gyrus (IFG), transcranial direct current stimulation (tDCS), verbal fluency

## Abstract

Individuals with attention deficit hyperactivity disorder (ADHD) struggle with impaired verbal fluency as an executive function. The left and right dorsolateral prefrontal cortex (dlPFC) and the right inferior frontal gurus (IFG), which show reduced functionality in individuals with ADHD, are involved in verbal fluency. In this study, a total of thirty-seven children with ADHD participated in two separate experiments. Each experiment included three different stimulation conditions: anodal left dlPFC/cathodal right vmPFC stimulation, the reversed montage, and a sham stimulation in Experiment 1, and anodal right dlPFC, anodal right IFG with extracranial return electrode, and a sham stimulation in Experiment 2. During each session, participants performed semantic and phonemic verbal fluency tasks while receiving tDCS. The results revealed a significant main effect of stimulation condition on phonemic verbal fluency during anodal left dlPFC stimulation in Experiment 1, and on semantic verbal fluency during both real stimulation conditions in Experiment 2. In conclusion, this study suggests that anodal left dlPFC stimulation improves phonemic verbal fluency, while anodal right dlPFC and right IFG stimulation enhance semantic verbal fluency. This domain-specific improvement can be attributed to the distinct cognitive demands of phonemic and semantic verbal fluency tasks. Phonemic verbal fluency heavily relies on working memory processes, whereas semantic verbal fluency requires effective inhibitory control and cognitive flexibility.

## 1. Introduction

Attention deficit hyperactivity disorder (ADHD) is a prevalent neurodevelopmental disorder characterized by two primary symptoms: inattention and hyperactivity–impulsivity [1]. Individuals affected by ADHD exhibit compromised cognitive processing in various domains, including perceptual [2,3], attentional [4], temporal [5], emotional [6], executive [7], social [8], and motivational [9] processing.

Two main conceptual models, the motivational theory [10] and the executive theory [7], are commonly used to explain the processing deficits observed in individuals with ADHD. The former emphasizes impaired hot executive functions, including delay aversion and reward processing, as the cognitive foundations for the behavioral symptoms associated with ADHD. The latter focuses on impaired cold executive functions (EFs), such as inhibitory control [11], cognitive flexibility [12], and working memory [13]. EFs refer to the ability to monitor and manipulate information to manage behaviors [14]. Three fundamental domains of EFs are inhibitory control, working memory, and cognitive flexibility [15,16]. In addition to the aforementioned deficits in these core EFs, children with ADHD also experience difficulties in other broader functions that are influenced by impaired EFs. These include challenges in emotion regulation [17], social skills [18], and verbal fluency [19].

Verbal fluency refers to the ability to retrieve words from memory in an organized manner [20]. It provides insights into verbal abilities, executive functions, and language production [21,22]. Verbal fluency encompasses two distinct types: semantic and phonemic. Semantic verbal fluency involves generating words within a specific semantic category, such as animals or fruits, within a given time limit of 60 s. On the other hand, phonemic verbal fluency requires participants to produce words that start with a designated letter [20].

Verbal fluency involves a variety of cognitive functions that contribute to its successful execution. These functions include working memory, which facilitates the recall and memorization of the generated items from long-term memory [23] Inhibition plays a crucial role in verbal fluency by suppressing irrelevant words, allowing for the production of relevant responses [24]. Cognitive flexibility is essential for verbal fluency as it enables individuals to shift between different response options, adapting to changing task requirements [25]. Attention is vital to maintain focus during verbal fluency tasks, ensuring efficient and rapid reporting of words [26]. Language skills are fundamental in verbal fluency as they enable individuals to vocalize their responses [27]. Lastly, long-term memory serves as the primary source of storage, allowing individuals to retrieve words from its network [28]. While verbal fluency encompasses a broad range of cognitive functions, its comprehensive nature offers the opportunity to assess various cognitive abilities through a simple and efficient test. Although identifying specific impaired cognitive functions may be challenging due to the diverse domains involved, the versatility of verbal fluency testing allows for a comprehensive evaluation of cognitive abilities. Verbal fluency has been recognized as a quick and efficient assessment tool for evaluating cognitive impairments in individuals with ADHD [29,30,31,32,33]. Furthermore, cognitive training interventions have been shown to effectively enhance verbal fluency in children with ADHD. Consequently, the sensitivity of this test to improvements in verbal fluency makes it a valuable tool for assessing cognitive progress in this particular group [34].

Transcranial direct current stimulation (tDCS) is a noninvasive brain stimulation technique that offers the potential to modulate the pathophysiological mechanisms underlying various psychopathological conditions by altering neural excitability [35,36]. The technique has the ability to modify neuronal resting membrane potentials, resulting in enhanced cortical excitability under the anode electrode and reduced excitability under the cathode electrode [36,37]. It can effectively modify the pattern of information processing in the brain, leading to improvements in cognitive functions [38,39]. Numerous tDCS studies found improved cognitive functions in children with ADHD. Improved working memory has been shown during anodal left dlPFC/cathodal right vmPFC [40,41] and anodal left/cathodal right dlPFC stimulation [41]. Improved inhibition has been described during anodal right dlPFC coupled with an extracranial reference electrode [42]; anodal right vmPFC combined either with an extracranial return electrode [43] or with cathodal left dorsolateral prefrontal [41,44]; cathodal left dlPFC/anodal right vmPFC [41,44]; and anodal right IFG with an extracranial return electrode [45]. Improved cognitive flexibility has been shown during anodal left dlPFC/cathodal right vmPFC [41]; for review, see [46,47,48]. However, a null effect of stimulation on working memory, inhibitory control, and cognitive flexibility has been described during anodal right and left dlPFC stimulation with extracranial return electrodes [49].

Numerous studies have explored the effects of tDCS on verbal fluency, with particular focus on the left IFG [50], right IFG [51], left dlPFC [52], and right dlPFC [53]. Among the main candidate regions, the left IFG, commonly known as Broca’s area, has been extensively investigated for its involvement in response selection, representing the verbal component of verbal fluency. On the other hand, the right IFG and the bilateral dorsolateral prefrontal cortex (dlPFC) have been attributed to the executive and fluency components of verbal fluency. Thus, the literature has highlighted distinct functional roles for these brain areas in contributing to the complexities of verbal fluency, prompting research endeavors to examine how tDCS modulation of these regions may impact verbal fluency performance.

In the present study, our hypothesis was centered around the presence of impaired verbal fluency in children with ADHD, the potential association of this impairment with other cognitive deficits, and the potential for cognitive function improvement through tDCS. The primary objective of this study was to assess the effects of tDCS on enhancing verbal fluency in children diagnosed with ADHD. To achieve this objective, we conducted two separate studies focusing on the modulation of specific brain regions. In the first study, we applied anodal and cathodal stimulation to the left dorsolateral prefrontal cortex (dlPFC), as it plays a crucial role in working memory. In the second study, our target was the right dlPFC and right inferior frontal gyrus, two key structures involved in inhibitory control and language production.

## 2. Materials and Methods

### 2.1. Participants

The present study consists of two independent experiments. Experiment 1 included 18 children with ADHD aged 6 to 12 (mean age: 8.83 ± 1.50). Experiment 2 recruited 19 children with ADHD aged 8 to 12 (mean age: 9.57 ± 1.38). All participants underwent a thorough assessment conducted by a professional child psychiatrist, resulting in a diagnosis of ADHD based on the criteria outlined in the *Diagnostic and Statistical Manual of Mental Disorders*, 5th edition [1]. Using the SNAP-IV scale, the participants in Experiment 1 had mild (*n* = 3), moderate (*n* = 19), or severe (*n* = 2) ADHD. Experiment 2 consisted of 7 participants with mild ADHD, 9 with moderate ADHD, and 3 with severe ADHD. Exclusion criteria included a history of traumatic brain injury (TBI), neurological disorders, or other significant psychiatric conditions. All participants had normal vision and were right-handed. It is worth noting that 9 participants in Experiment 1 and all of the participants in Experiment 2 were taking medication, such as methylphenidate and risperidone, during the study. These participants were instructed to discontinue their medication at least 12 h prior to the experimental sessions. Table 1 displays the demographic characteristics of the participants. This study adhered to the ethical guidelines outlined in the Helsinki Declaration of 1975, as revised in 2013, and received approval from the ethical committee of Shahid Beheshti University.

### 2.2. Swanson, Nolan, and Pelham Scale (SNAP-IV)

The SNAP-IV scale is based on the diagnostic criteria of ADHD outlined in the DSM-IV [54]. It comprises 18 items that assess symptoms of inattention, hyperactivity, and impulsivity. Caretakers or teachers rate the severity of these symptoms on a four-point Likert scale, reflecting the symptomatology experienced by the child over the previous four weeks. In Iran, the SNAP-IV scale has been validated, and Cronbach’s alpha coefficients of 0.81 for hyperactivity/impulsivity and 0.75 for inattention have been reported [55].

### 2.3. Semantic Verbal Fluency Test (SVFT)

The SVFT assesses the spontaneous production of words within specific search parameters, focusing on verbal association fluency. Participants are instructed to generate as many exemplars as possible from a given semantic category, commonly animals or fruits, within a time limit of 60 s [20]. In the present study, participants were asked to generate items from the animals and fruits categories.

### 2.4. Phonemic Verbal Fluency Test (PVFT)

The PVFT evaluates the ability to generate words starting with a particular letter, often F, A, or S, within a designated time frame [20]. In our study, participants were instructed to generate words beginning with the Persian letters /be/ and /fe/, which correspond to the English letters B and F, respectively, within 60 s.

### 2.5. Protocols for the tDCS

In this study, tDCS was administered using a battery-driven stimulator (ActivaTek Inc., Gilroy, CA, USA) with a pair of saline-soaked sponge rubber electrodes measuring 5 cm × 5 cm. The electrode placements followed the 10–20 EEG system. In Experiment 1, the tDCS montages consisted of two configurations: (1) anodal stimulation over the left dlPFC (F3) coupled with cathodal stimulation over the right vmPFC (Fp2), and (2) anodal stimulation over the right vmPFC (Fp2) coupled with cathodal stimulation over the left dlPFC (F3). In Experiment 2, the electrode arrangements were as follows: (1) anodal stimulation over the right dlPFC (F4) with the reference electrode on the contralateral arm, and (2) anodal stimulation over the right inferior frontal gyrus (F8) with the reference electrode on the contralateral arm. Each experiment included three sessions of tDCS stimulation. In addition, sham stimulation was administered as a control condition. During real stimulation, the current intensity was set at 2 mA and lasted for 15 min, with 30 s ramp-up and -down periods. In both experiments, during the sham condition, the electrode placements precisely matched one of the real conditions, with random allocation across participants. However, in the sham conditions, the stimulator was discreetly deactivated after a brief 30 s ramp-up period, ensuring that participants remained unaware of the stimulation’s absence. This sham procedure was implemented to provide a reliable control for potential placebo effects, allowing for a rigorous evaluation of the true impact of transcranial stimulation in the experimental settings. This sham stimulation method, commonly used in tDCS studies, does not induce long-term effects on cortical excitability and is suitable for blinding purposes [56,57].

### 2.6. Experimental Procedures

At the beginning of the experiments, participants’ caretakers provided informed consent by signing a consent form. The study employed a single-blinded crossover design, and to enhance statistical power, a sham-controlled within-subject design was utilized, with participants acting as their own controls. To minimize potential carry-over effects, a one-week interval was implemented between sessions. The order of stimulation montages was counterbalanced across participants in each experiment. Each tDCS session lasted for 20 min and was followed by the completion of the verbal fluency task. In each session, the tasks commenced five minutes after the initiation of stimulation, Figure 1. During the verbal fluency tasks, the researcher presented the instructions to the participants, followed by a request to promptly and accurately repeat as many words related to the given instruction as possible. Throughout these tasks, the participant’s voice was diligently recorded for subsequent processing and analysis. This systematic approach allowed for comprehensive examination of verbal fluency performance and contributed to the overall data collection process. It is important to note that the study also included other tasks, including emotion regulation tasks, which have been reported in separate publications. Two researchers/authors (R.E. and Z.H.) conducted the tDCS sessions and supervised the behavioral tasks. At the end of each session, participants completed a side-effect checklist following the guidelines outlined by [58]. Additionally, participants were asked to guess whether they had received real or sham stimulation. In this study, the researchers employed a rigorous control technique known as a sham-controlled within-subject design to investigate the effects of tDCS on participants’ performance in a verbal fluency task. The study utilized a single-blinded crossover design, in which participants acted as their own controls, undergoing both the experimental condition and the sham condition. This design ensured that any observed effects were not solely attributable to the participants’ expectations or beliefs about the stimulation. Figure 2 depicts the current distribution in the cortical structures.

### 2.7. Data Analysis

In the current study, data analysis was performed using IBM SPSS Statistics 23. The normality of the data was confirmed using the Kolmogorov–Smirnov test. To investigate the effect of stimulation (with three different montages in each experiment) on task performance, a one-factor repeated measures analysis of variance (one-way ANOVA) was conducted. The dependent variables were the “semantic” and “phonemic” scores of the verbal fluency task. Data sphericity was assessed using Mauchly’s test of sphericity, and if necessary, the degrees of freedom were adjusted using the Greenhouse–Geisser test. Additionally, post hoc analyses were performed using Fisher’s LSD test in the case of significant outputs from the respective ANOVAs. Furthermore, session order and medication use were included as covariates in an analysis of covariance (ANCOVA). A significance level of 0.05 was used for all statistical comparisons.

## 3. Results

The participants in both experiments tolerated tDCS well, with no reported serious side effects. Some participants reported mild and tolerable sensations of itching, tingling, and burning beneath the electrodes at the beginning of stimulation. The descriptive statistics for side effects in different stimulation conditions are presented in Table 2. In Experiment 1, there were no significant differences between conditions for most side effects, except for burning sensation (F_2_ = 5.04, *p* = 0.01, ηp2 = 0.24). LSD post hoc analysis revealed that participants reported a higher level of burning during anodal dlPFC/cathodal vmPFC (MD = 0.72, *p* = 0.01) and reversed electrode positions (MD = 0.88, *p* = 0.02) compared to the sham stimulation condition. In Experiment 2, participants reported a higher sense of pain during anodal dlPFC (F_2_ = 3.42, *p* = 0.04, ηp2 = 0.16; MD = 0.89, *p* = 0.01) and anodal right IFG (F2 = 3.42, *p* = 0.04, ηp2 = 0.16; MD = 0.68, *p* = 0.05) compared to the sham condition. Additionally, compared to the sham condition, anodal dlPFC and anodal right IFG induced a higher sensation of tingling (F_2_ = 7.06, *p* < 0.01, ηp2 = 0.28; MD = 1.15, *p* < 0.01) and (F_2_ = 7.06, *p* < 0.01, ηp2 = 0.28; MD = 0.63, *p* = 0.04), respectively.

Regarding verbal fluency, mean scores, standard deviations, and ANOVA results for the phonemic and semantic blocks of the task in each stimulation condition are presented in Table 3. In the first experiment, there was a significant effect of stimulation on the phonemic subscale of the task (F_1.23_ = 4.63, *p* = 0.03, ηp2 = 0.21). LSD post hoc analysis revealed that anodal left dlPFC/cathodal right vmPFC led to higher scores on the phonemic subscale compared to the sham condition (MD = 3.11, *p* < 0.001). However, there was no significant effect of stimulation on the semantic subscale (F_2_ = 1.15, *p* = 0.32, ηp2 = 0.06). In Experiment 2, the ANOVA showed a significant effect of stimulation type on the semantic block of the task (F_2_ = 3.97, *p* = 0.02, ηp2 = 0.18). Paired comparisons revealed that, compared to the sham stimulation condition, participants achieved higher scores in both anodal right dlPFC (MD = 3.36, *p* = 0.04) and anodal right IFG (MD = 3.05, *p* = 0.02), as shown in Figure 3.

## 4. Discussion

The results of the study revealed a significant improvement in both semantic and phonemic verbal fluency during stimulation. However, this improvement was specific to certain brain regions, with the left dlPFC being involved in phonemic verbal fluency enhancement, and the right vmPFC and right IFG being associated with improved semantic fluency. In the subsequent discussion, we will delve into a detailed analysis of our findings, considering the causal role of these specific brain areas in relation to their respective contributions to verbal fluency.

### 4.1. Improved Phonemic Verbal Fluency

Regarding phonemic verbal fluency, the enhancement was observed exclusively in Experiment 1, which targeted the left dlPFC. During anodal left dlPFC stimulation, we observed a significant improvement in phonemic verbal fluency compared to the sham stimulation condition. However, no such effect was found for semantic verbal fluency. Notably, cathodal left dlPFC stimulation did not have any detrimental effect on either phonemic or semantic verbal fluency. These findings suggest that the left dlPFC plays a critical role in phonemic verbal fluency tasks. The observed improvement in phonemic verbal fluency can be interpreted within the context of the involvement of the dlPFC in working memory and cognitive flexibility processes. Earlier tDCS studies found improved working memory performance in children with ADHD during anodal dlPFC stimulation with both cathodal left dlPFC [59] and cathodal right vmPFC [40,60]. Improved cognitive flexibility has also been shown during anodal left dlPFC/cathodal right vmPFC [59]. Improved phonemic fluency through anodal left dlPFC stimulation has previously been described in healthy adults [52] and in adults with autism spectrum disorder [61], Parkinson disease [62,63], and post-stroke aphasia [64].

### 4.2. Improved Semantic Verbal Fluency

Improved semantic verbal fluency was observed in Experiment 2, which targeted the right dlPFC and the right IFG. These two brain structures have been identified as key players in inhibitory control processes in individuals with ADHD. Notably, meta-analyses of neuroimaging studies have consistently revealed functional abnormalities in inhibitory control networks, specifically the right-hemispheric fronto-basal-ganglia networks, including the right IFG and striatal regions [65,66,67]. Earlier tDCS studies found improved inhibitory control in children with ADHD during anodal right dlPFC stimulation coupled with both cathodal left dlPFC stimulation [44] and an extracranial return electrode [42]. Improved verbal creativity has been found during anodal right/cathodal left IFG stimulation [51]. Experiment 1 did not show improvement in semantic verbal fluency, indicating that the left dlPFC has no role in semantic verbal fluency. An earlier study in healthy adults found decreased semantic verbal fluency during anodal left dlPFC coupled with cathodal vmPFC [68].

Functional neuroimaging studies have consistently demonstrated the importance of the frontal lobe in phonemic fluency tasks and the relevance of the temporal lobe in semantic fluency tasks [69,70,71].

### 4.3. Domain-Specific Improvement

The results revealed that the left dlPFC, right dlPFC, and right IFG exhibited domain-specific effects on verbal fluency. Upregulation of the left dlPFC resulted in improved phonemic verbal fluency, whereas upregulation of the right dlPFC and right IFG led to enhanced semantic verbal fluency. This domain-specific effect can be interpreted in the context of the distinct cognitive components involved in these two types of verbal fluency tasks. While phonemic verbal fluency and semantic verbal fluency share many cognitive processes, such as sustaining attention, devising search strategies, selecting appropriate words, inhibiting competitors, engaging working memory, and articulating output, there are notable distinctions between them. Phonemic fluency relies on selecting and retrieving information based on spelling (orthography), while semantic fluency places a greater demand on conceptual knowledge stores. Furthermore, semantic verbal fluency requires more inhibition and shifting, as participants need to generate words from within a specific category and then switch to a different category. In contrast, phonemic verbal fluency relies more heavily on working memory, as participants are required to recall words starting with specific phonemes from various categories. Indeed, inhibiting semantically relevant information is more challenging than inhibiting semantically irrelevant information. This phenomenon is often referred to as semantic interference. When performing tasks that require the suppression of related or semantically similar items, individuals may experience greater difficulty due to the inherent interconnectedness of the semantic network in the brain [72]. By elucidating the domain-specific effects of tDCS on verbal fluency, this study provides insights into the underlying cognitive processes and neural mechanisms involved in these tasks. Understanding the distinct contributions of the left dlPFC, right dlPFC, and right IFG to phonemic and semantic fluency enhances our knowledge of the specific cognitive domains targeted by tDCS interventions and has implications for developing more targeted and effective interventions for individuals with ADHD.

In this study, the order of sessions/tests was found to be nonsignificant when treated as a covariate, indicating that the sequence in which participants underwent the tDCS sessions and verbal fluency tests did not significantly influence the observed results. This finding suggests that any potential learning effect on the verbal fluency task was unlikely to be a confounding factor, particularly after the one-week interval between sessions. The absence of a significant order effect strengthens the internal validity of the study and provides greater confidence in attributing the observed changes in verbal fluency performance to the effects of tDCS rather than a learning or practice effect. By ruling out a learning effect as a major contributor, the study’s outcomes offer more compelling evidence for the potential cognitive benefits of tDCS in enhancing verbal fluency, supporting the validity of the reported findings and their implications for memory-based operations.

In conclusion, the findings of this study indicate that anodal left dlPFC stimulation leads to improved phonemic fluency, while anodal right dlPFC and right IFG stimulation result in enhanced semantic verbal fluency. This domain-specific improvement can be attributed to the distinct cognitive demands of phonemic and semantic verbal fluency tasks. Phonemic verbal fluency relies heavily on working memory processes, whereas semantic verbal fluency requires effective inhibitory control and cognitive flexibility. These results contribute to our understanding of the neural mechanisms underlying verbal fluency and highlight the potential of tDCS as a modulatory tool for enhancing specific aspects of verbal fluency in children with ADHD.

### 4.4. Limitations and Future Directions

Several limitations should be considered when interpreting the findings of the present study. First, the study focused on investigating the impact of tDCS on verbal fluency in three specific brain areas. However, it is important to acknowledge that verbal fluency involves a network of neural structures, including areas such as the posterior parietal cortex [52] and the temporo-parietal junction [73]. Future studies could explore the involvement of these additional brain regions to gain a more comprehensive understanding of the neural basis of verbal fluency. Second, the present study employed a single-blind design with single-session interventions. While this design allowed for initial exploration of the effects of tDCS on verbal fluency, it is crucial to consider the clinical implications cautiously. Future studies should incorporate multisession designs to assess the long-term effects of tDCS on verbal fluency in children with ADHD. This would provide more robust evidence regarding the potential clinical application of tDCS for improving verbal fluency in this population. Furthermore, the study utilized similar tests across different sessions to evaluate verbal fluency performance. This limitation was unavoidable due to the variations in word frequency and magnitude across different lexicons, leading to differences in difficulty levels among categories. Although the study implemented a one-week intersession interval and found no significant effect of session order as a covariate, the use of parallel tests would have been beneficial in minimizing potential learning effects. For this purpose, piloting a behavioral study and creating balanced versions of the verbal fluency test with different categories and letters could have further strengthened the experimental design. In the present study, we investigated verbal fluency in a sample of children with ADHD and executive impairments. The verbal fluency task, which comprised both language and executive components, allowed us to examine the interplay between these cognitive processes in this specific population. However, it is crucial to acknowledge that our research design and targeted brain areas primarily focused on executive functions, limiting the scope for a comprehensive discussion of the language component of the task. Considering the heightened pain perception observed in real conditions as opposed to sham conditions, it appears that the blinding might not have been entirely successful. This warrants a degree of caution in the interpretation of the results. Another limitation of this study is the absence of a healthy control group for comparison. As our research focused solely on a group of ADHD children with executive impairments, we were unable to fully explore the influence of the pathological model on the observed results and its relation to verbal fluency performance. In conclusion, while the present study contributes valuable insights into the effects of tDCS on verbal fluency in children with ADHD, it is important to consider the aforementioned limitations. Future research should address these limitations by including a wider range of brain regions, employing multisession designs, and utilizing parallel tests to further advance our understanding of tDCS as a potential intervention for improving verbal fluency in clinical settings.

## Figures and Tables

**Figure 1 brainsci-13-01257-f001:**
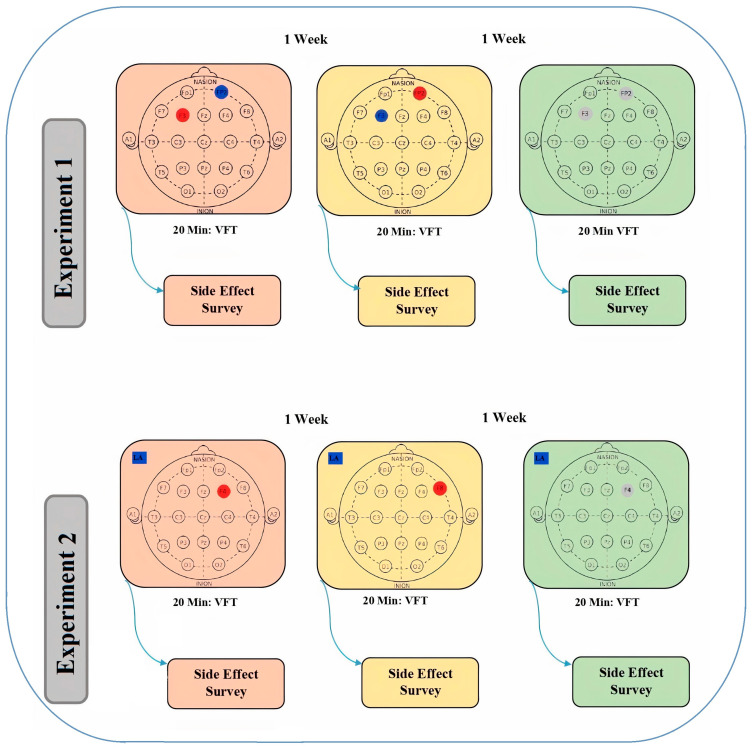
Experimental procedures. In each study, participants received one of the tDCS protocols in randomized order. The VFT was performed in each session 5 min after stimulation, and the final step was completion of the side-effect questionnaire. Note: VFT: verbal fluency test; LA: left arm.

**Figure 2 brainsci-13-01257-f002:**
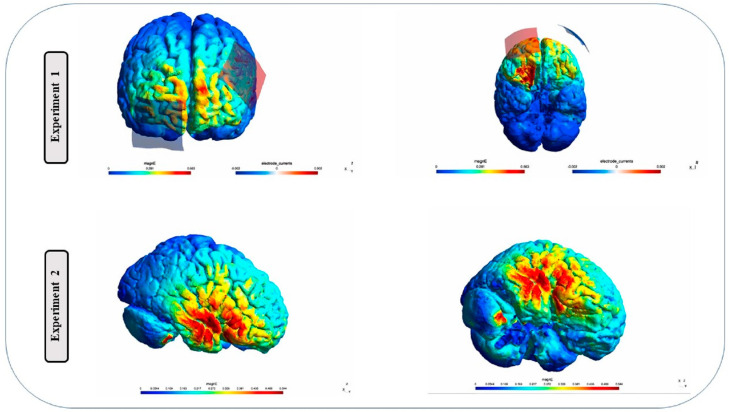
Distribution of electrical field calculated using SimNIBS. Experiment 1: two 5 × 5 cm electrodes were positioned over Fp2 and F3, and the current intensity was set to 2 mA. Experiment 2: two 5 × 5 cm electrodes were positioned over F4 or F8, and the current intensity was set to 2 mA.

**Figure 3 brainsci-13-01257-f003:**
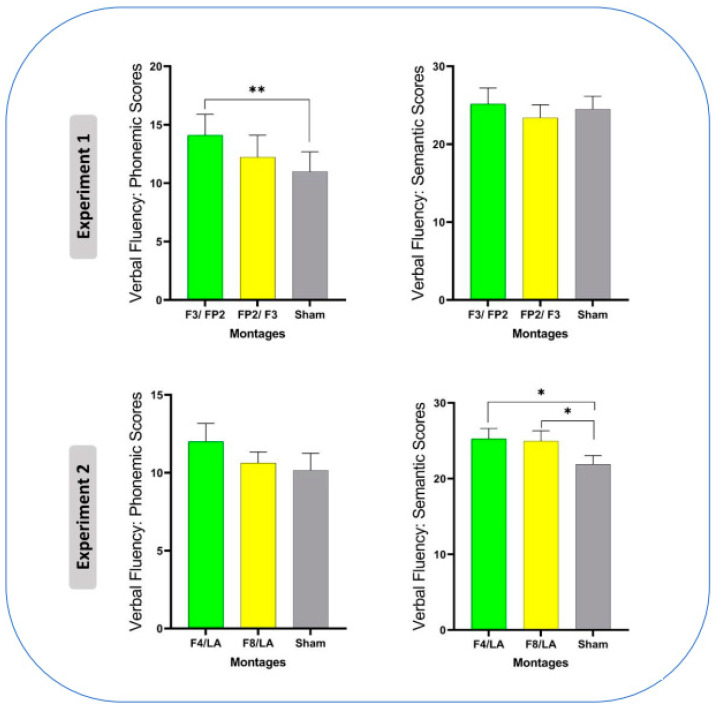
Shown are the effects of tDCS on the outcome measures. The vertical axis indicates the respective outcome measures of the tasks. The bars show the means, and the error bars represent standard error of means. The horizontal axes show stimulation conditions. The electrode placements followed the 10–20 EEG system, with the abbreviations before and after the slash indicating anodal and cathodal electrode placement, respectively. LA: left arm. *: *p* < 0.05, **: *p* < 0.001.

**Table 1 brainsci-13-01257-t001:** Demographic characteristics and ADHD rating of participants.

Variables	M (SD)	
	Experiment 1	Experiment 2
Age (Years)	8.83 (1.50)	9.57 (1.38)
Education (Years *)	2.89 (1.45)	3.42 (1.60)
Gender (Male/Female)	14/4	18/1
SNAP-IV	29.04 (8.63)	23.32 (11.41)

Abbreviations: M: mean; SD: standard deviation; F: female; M: male. * The number of academic years a person completed in a formal program.

**Table 2 brainsci-13-01257-t002:** Descriptive statistics for tDCS side effects in the different stimulation conditions and the results of the respective ANOVAs.

	Conditions, M (SD)			Statistics			
Experiment 1	F3/FP2	FP2/F3	Sham	df	F	*p*	ηp2
Pain	0.11 (0.32)	0.06 (0.23)	0.00 (0.00)	1.45	1.00	0.35	0.05
Vertigo	0.00 (0.00)	0.00 (0.00)	0.00 (0.00)	2			
Burning	1.17 (1.04)	1.33 (1.28)	0.44 (0.61)	2	5.40	0.01	0.24
Tingling	0.61 (0.77)	0.89 (1.02)	0.56 (0.92)	2	1.11	0.34	0.06
Confusion	0.00 (0.00)	0.00 (0.00)	0.00 (0.00)	2			
Drowsiness	0.33 (0.76)	0.06 (0.23)	0.22 (0.73)	2	1.12	0.33	0.06
Experiment 2	F4/LA	F8/LA	Sham	df	F	*p*	ηp2
Pain	1.11 (1.76)	0.89 (1.41)	0.21 (0.71)	2	3.42	0.04	0.16
Vertigo	0.16 (0.37)	0.32 (0.74)	0.05 (0.22)	1.47	1.28	0.28	0.06
Burning	1.00 (1.82)	0.89 (1.24)	0.37 (1.16)	1.47	1.28	0.28	0.06
Tingling	1.42 (1.80)	0.89 (1.32)	0.26 (0.73)	2	7.06	<0.01	0.28
Confusion	0.53 (0.90)	0.63 (1.11)	0.11 (0.45)	1.25	3.68	0.05	0.17
Drowsiness	0.74 (1.28)	0.58 (0.90)	0.37 (1.16)	2	0.53	0.59	0.02

Abbreviations: M: mean; SD: standard deviation; LA: left arm; df: degrees of freedom; F: F-value, *p*: *p*-value; ηp2: partial eta squared. The electrode placements followed the 10–20 EEG system, with the abbreviations before and after the slash indicating anodal and cathodal electrode placement, respectively.

**Table 3 brainsci-13-01257-t003:** Descriptive statistics and the results of one-way ANOVAs for the study measures.

	Conditions, M (SD)			Statistics			
Experiment 1	F3/FP2	FP2/F3	Sham	df	F	*p*	ηp2
Phonemic	14.11 (7.59)	12.22 (7.98)	11.00 (7.12)	1.23	4.63	0.03	0.21
Semantic	25.17 (7.80)	23.39 (7.04)	24.50 (6.99)	2	1.15	0.32	0.06
Experiment 2	F4/LA	F8/LA	Sham	df	F	*p*	ηp2
Phonemic	12.00 (5.09)	10.63 (3.04)	10.16 (4.74)	2	1.50	0.23	0.07
Semantic	25.26 (5.94)	24.95 (5.08)	21.89 (5.01)	2	3.97	0.02	0.18

Abbreviations: M: mean; SD: standard deviation; LA: left arm; df: degrees of freedom; F: F-value; *p*: *p*-value; ηp2: partial eta squared. The electrode placements followed the 10–20 EEG system, with the abbreviations before and after the slash indicating anodal and cathodal electrode placement, respectively.

## Data Availability

Data for this study are available upon request.

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
