# Peer review of "Transcranial Direct-Current Stimulation Improves Verbal Fluency in Children with Attention Deficit Hyperactivity Disorder (ADHD)"

_brainsci, 2023, doi:10.3390/brainsci13091257_

Round 1

Reviewer 1 Report

Thanks for submitting the work. The experiments conducted in the work showed that anodal left dlPFC stimulation improved phonemic verbal fluency, while anodal right dlPFC and right IFG stimulation improved semantic verbal fluency. The authors argue that tDCS can modulate specific aspects of verbal fluency by targeting distinct cognitive processes and neural mechanisms. The introduction to the work is sufficient, and the methods are well documented. Also, the flow of the text in the article is very nice.

However, I have a few suggestions that could strengthen your work:

  1. The abstract should not be divided into specific sections. Rather the different sections should be integrated into one paragraph.

  2. Please add a figure to the methods section that shows the schematic diagrams of the stimulation montages. This would help readers visualize the electrode placements and schemes.

  3. Please provide a detailed explanation of how the stimulation current and duration were determined.

  4. Please fix the formatting of the second reference in line 168.

  5. Please elaborate on the control technique, in section 2.6. 

  6. Since other tests were also conducted following the stimulation, they should be mentioned here. It would allow the reader to assess any effect those might have on the current study.

  7. Also, it would be interesting to know how the observed effects on verbal fluency depends on the time interval between the stimulation and the tests. The authors should show data, or provide insights on this. Although this has been briefly mentioned in the discussion, it deserves more attention, as it is imperative for memory based operations.

Author Response

Dear Reviewer

Thank you for providing us with the opportunity to revise the manuscript. We are pleased to have received valuable and informative comments from the expert reviewers. In this point- by-point letter, we explain all changes based on the comments. Additionally, to facilitate easy identification of the modifications, our responses to the comments will be written in blue text. We have highlighted the added sections and used strikethrough formatting for the deleted parts.

All the best,

The author

Reviewer 2 Report

Nejati and colleagues describe a series of experiments in which the role of frontal lobes in verbal fluency for ADHD patients is tested. The paper is well written even if a good amount of background is missing, and I have some comments to address before it can be considered for publication.

Introduction:

Line 37-38 “The former emphasizes impaired non-executive 37 functions as the cognitive foundations” please expand which non-executive functions are involved.

Lines 82-83: please refer to anodal electrode and cathodal electrode as anode and cathode

Line 108: Concerning verbal fluency, a key region is the LEFT IFG, rather than the RIGHT one. As this region (rIFG) may be crucial for inhibitory reasons, the lIFG is crucial for response selection and has been the target of many previous works with TMS and tDCS aiming at modifying verbal fluency performance (e.g. Cattaneo et al., 2010; Holland et al., 2011). Furthermore, imaging evidence showed that activity in this region is associated with task performance (Meinzer et al., 2013). Finally, tDCS modulation acted through a functional network involving left premotor-IFG, enhancing its activity, which correlated with task performance (Pisoni et al., 2018). Please better address the existing literature and why these regions were chosen and how this potentially affected the results.

Methods

No rationale is given for the different montages. Furthermore, a modeling of the electrical current spread with these montages could help in discussing which regions were mostly affected by the stimulation protocol.

It would be interesting running a 2x3 ANOVA to see also if verbal and semantic fluency show differences between them according to the stimulation protocol. 

A picture of the montages and of the current modeling could improve the clarity of the procedures. 

Results

Figure 1 needs a differentiation between panels and a description to understand what these barplots refer to.

No information is reported for the medication status even if this was included in the ANCOVA. The same holds for stimulation session.

Categories and letters were always the same. Can the authors rule out a learning effect by showing data of the first sessions only? Furthermore, by piloting a behavioral study, authors could have created and balanced different categories and letter to create parallel versions of the test (e.g. see Pisoni et al., 2018)

Discussion

Lines 244-261: the lDLPFC is also a region which is found active during the task, this should be included in the comments. Furthermore, references are missing for the authors’ claims.

Paragraph 4.2 may need a deep rework after some more references in both imaging studies and neuromodulation of the left hemisphere during verbal fluency, since many studies found the left dlpfc linked to verbal fluency and semantic fluency also. Please update and complete the framework. Furthermore, a modelling of current spread is needed to comment with this specificity your results. 

Paragraph 4.3 and in general the whole paper, completely neglect literature on how linguistic characteristics and language organization impact on verbal fluency performance, and misread some of it. First, both phonemic and semantic fluency rely on clustering and switching (see Binney et al., 2018). So the cluster has to me anchored (being it semantic or phonemic) and then, when it has been emptied, a switch towards the following one must occur to proficiently perform the task. Semantic fluency, in this regard, involves less switching, and a more naturalistic exploration of the cluster, as normally language is organized in semantic networks. Please take into account not only executive functions, but also language when you are commenting on a language task. It is true, as the authors said, that phonemic fluency is more demanding, but I would change the term “working memory” to “Executive functions”, since it is not only a mere question of memory. The Treisman theory do not fit here, since the engagement and disengagement that the authors cite do not occur during semantic memory, where activation spreads between nodes of the semantic networks more naturally, rending the task easier as it is usually reported in the literature as compared to the phonemic one. Also the Baddeley’s model of working memory has little to do with the cognitive processes going on during phonemic fluency. Or the authors better describe the paragraph or please leave this citation out.

The author do not comment the fact that perceptual aspects of the stimulation were differently reported by participants. This indicates that the blinding procedures were not optimal. Please comment and potentiali tone down the claims. 

Finally, little to nothing is discussed here according to how the pathological model under investigation might have affected the results, or relates to them. 

Ethics approval and consent to participate: 

This part is missing, please complete. 

N/A

Author Response

(The authors gave the same response as above.)

Reviewer 3 Report

Nejati and colleagues presented a study on how tDCS can improve verbal fluency in children with ADHD. While this is not entirely novel, studies like this one sheds more like on specific areas, and parameters needed to identify and improve brain areas associated with cognitive functions. I applaud the authors for carrying out this study. The paper was well written; however, I have some concerns:

Major concerns

1.    Please clarify the montage for experiment 1: Lines 154-57. Were there two active electrodes at the right and left DLPFC respectively?. As in, anodal at left and reference somewhere else, and then cathodal at right and reference somewhere else?. Are you referring to bilateral montage at right DLPFC where anodal = active electrode; and cathodal = reference electrode at left DLPFC?. If you use anodal and cathodal to refer to a bilateral or frontal/CSO montages, you must be very clear. Preferably, a diagram would be nice. The explanation in experiment 2 is good.

2.    Please explain what is written in the table briefly under the table. All table descriptions are missing. Please do this for the figures as well. I had a hard time understanding the figures without a good description. For example what are the colored bars for?. Are the bar graphs at the top for experiment 1, and those at the bottom for expt 2?. What is F4/A?. In the methods, there was no description of the sham montage. Did each montage have their own sham configuration?.

3.    The study procedure leaves much to be desired. There was no clear description of the experimental procedure. How the tasks were performed, and how stimulation was done during the experiment are missing. Were the tasks performed during tDCS or before/after?. What is the timeline of the experiment?. A diagram showing the beginning to the end will be better.

Minor concerns

Please check/edit the entire manuscript for grammar, and punctuation. I saw a few issues with language, but I am not sure I caught all language errors.

1.    Line 78. Please always start sentence with capital letter. ‘Transcranial’ nor ‘transcranial’.

2.    Line 81. Please check the in-text citation to harmonize the bibliography style. I believe the citation Michael A Nitche et al., 2008 should rather be Nitsche et al., 2008, just like the other citations. Please check this through the whole manuscript. Another example: Line 256-57, Nejati et al., 2017, and V. Nejati et al., 2017.

3.    Lines 122-124, the sentence should be re-written to include a ‘,’, and ‘and’.

ð  ‘It is worth noting that 9 participants in Experiment 1, and all of the participants in Experiment 2 were taking medication during the study, such as methylphenidate and risperidone’.

4.    Line 236: The subsection 4 should not be called ‘conclusion’ but rather ‘discussion’.

5.    Table 1. Error: Yearsa => Years.

6.    Font size differences in the abstract.

Minor editing for language especially punctuation, and grammatical errors is very much recommended. This also includes citation style inconsistencies. 

Author Response

(The authors gave the same response as above.)

Round 2

Reviewer 3 Report

I appreciate very much the efforts of the authors in proving a response to my comments, and I accept almost every answer except the about how to write tDCS at the beginning of a sentence.

I disagree with the authors' response/argument on this issue (Line #70).

In English, all nouns are capitalized at the beginning of a sentence. In rare exceptions, acronyms are written as they are. In this particular paper, you can always write tDCS at the beginning of a sentence without capitalizing the first letter (that is the convention). However, it is not grammatically correct to write 'transcranial' with a small 't' when beginning a sentence, more so at the beginning of a paragraph. 

Here is an example:

'Transcranial direct current stimulation (tDCS) is a non-invasive brain stimulation technique that uses weak direct currents to modulate brain physiology. tDCS has received a huge interest over the last two decades due partly to its simplicity in application, and its efficient neuromodulatory effects.'

In this example above, a paragraph is started with the word 'Transcranial'. The second sentence starts with a small 't' in tDCS. It is grammatically correct to write acronyms such as tDCS without capitalizing the first letter of the acronym. However what is not correct is writing 'transcranial' at the beginning of a sentence with a small 't'. 

I believe if some authors have gone away with this simple grammatical error in the past, it is not sufficient to make this a standard. There are numerous other papers with the correct grammar of how tDCS should be written. 

English is mostly fine except one minor issue with capitalizing the first letter of a word at the beginning of a sentence. 

Line 70.

Author Response

Dear Reviewer,

Thank you for dedicating your time and expertise to review our manuscript. We highly value your input, which has significantly contributed to enhancing the quality of our paper. We are pleased to accept your suggestion regarding the capitalization of 'tDCS' at the beginning of a sentence.

Your insightful explanation regarding English grammar conventions and the correct capitalization of acronyms is well-received. Upon careful consideration, we acknowledge the validity of your point and the importance of maintaining consistency in adhering to established grammatical standards. As such, we will revise our manuscript to ensure that 'tDCS' is capitalized appropriately at the beginning of sentences and paragraphs.

We sincerely appreciate your diligence in helping us improve the clarity and accuracy of our paper. Your feedback has been invaluable, and we are grateful for your commitment to the advancement of academic discourse.

Thank you once again for your meticulous review, and we look forward to incorporating your valuable insights into our manuscript.

Best regards,
The Authors